# *Fusobacterium nucleatum* Is Associated with Tumor Characteristics, Immune Microenvironment, and Survival in Appendiceal Cancer

**DOI:** 10.3390/microorganisms13071644

**Published:** 2025-07-11

**Authors:** Christopher Sherry, Neda Dadgar, Hyun Park, Chelsea Knotts, Erin Grayhack, Rose Blodgett, Kunhong Xiao, Ashten N. Omstead, Albert D. Donnenberg, David L. Bartlett, Vera Donnenberg, Ajay Goel, Ali H. Zaidi, Patrick L. Wagner

**Affiliations:** 1Allegheny Health Network Cancer Institute, Pittsburgh, PA 15224, USA; christopher.sherry2@ahn.org (C.S.); hyunyoung.park@ahn.org (H.P.); knotts.chelsea@ahn.org (C.K.); erin.grayhack@ahn.org (E.G.); rose.blodgett@ahn.org (R.B.); kevin.kh.xiao@gmail.com (K.X.); ashten.omstead@ahn.org (A.N.O.); albert.donnenberg@ahn.org (A.D.D.); david.bartlett@ahn.org (D.L.B.); vera.donnenberg@ahn.org (V.D.); ali.zaidi@ahn.org (A.H.Z.); 2Translational Hematology and Medical Oncology Department, Cleveland Clinic, Cleveland, OH 44195, USA; dadgarn@ccf.org; 3Department of Cardiothoracic Surgery, University of Pittsburgh, Pittsburgh, PA 15260, USA; 4Department of Medicine, Drexel University School of Medicine, Philadelphia, PA 19104, USA; 5City of Hope, Duarte, CA 91010, USA; ajgoel@coh.org

**Keywords:** appendiceal cancer, *Fusobacterium nucleatum*, immune microenvironment, tumor microbiome, survival outcomes

## Abstract

Emerging evidence highlights the role of the tumor microbiome, including *Fusobacterium nucleatum* (Fn), in a wide range of gastrointestinal cancers. Fn purportedly contributes to tumorigenesis by activating oncogenic pathways and modulating immune responses. Although the prevalence and impact of Fn has been extensively studied in colorectal cancer, no previous systematic or in situ studies have been performed in appendiceal cancer (AC). The aim of this study was to evaluate the prevalence and association of Fn density in AC with clinical factors and oncologic outcomes. Archival tissue from 54 patients with AC was assessed for Fn density using RNA in situ hybridization. Clinicopathological variables were obtained for each case through electronic medical record review, and the immune microenvironment was characterized in each case using immunohistochemistry to quantify CD3+ and CD8+ T lymphocytes and M1-/M2-like tumor-associated macrophages. In AC, Fn density was associated with patient age, tumor grade, and histologic subtype. Fn was negatively associated with CD3+ and CD8+ T lymphocytes and positively associated with M2-like TAMs in low-grade AC. Interestingly, tumor Fn content was associated with better overall and progression-free survival, even when controlling for tumor grade. In this exploratory study, we found that Fn is prevalent in AC. Fn is associated with a number of clinical, pathologic, immunologic, and prognostic variables in AC that are distinct from the corresponding observed associations in colorectal cancer. Further research is warranted to validate these findings and explore the mechanistic contributions of Fn to AC pathogenesis or immune response.

## 1. Introduction

Appendiceal cancer (AC) is a rare malignancy with diverse histological subtypes, including low-grade appendiceal mucinous neoplasms (LAMNs), mucinous adenocarcinomas (mACs), goblet cell adenocarcinomas (gcACs), and signet ring cell adenocarcinomas (srcACs) [1]. Unlike other gastrointestinal malignancies, AC is often diagnosed incidentally during appendectomy or presents at a late stage with extensive peritoneal spread. Progress in understanding the pathogenesis of AC has been hindered by its low incidence, and, as a result, treatment strategies have been largely extrapolated from those for colorectal cancer (CRC). However, an emerging understanding of the distinct biology of AC has motivated investigators to identify disease-specific mechanisms and potential therapeutic targets [2,3,4,5].

In recent years, increasing attention has been given to the role of the tumor microbiome in cancer development and progression. Among microbial agents, *Fusobacterium nucleatum* (Fn), a Gram-negative anaerobic bacterium, has emerged as a significant factor in gastrointestinal malignancies, particularly CRC [6]. Fn colonizes the tumor microenvironment through its adhesive and invasive properties, contributing to tumorigenesis via multiple mechanisms. These include the promotion of chronic inflammation, activation of pro-oncogenic pathways (e.g., β-catenin and NF-κB), and modulation of the immune response, enabling tumor immune evasion [7]. In CRC, Fn has been shown to suppress anti-tumor immunity by inhibiting the infiltration and function of CD8+ T lymphocytes, which are critical for recognizing and eliminating tumor cells [8]. Simultaneously, Fn fosters an immunosuppressive microenvironment by promoting the polarization of M2-like macrophages, which secrete anti-inflammatory cytokines and promote Tregs and a Th2 response [9]. These changes may impact the tumor immune microenvironment, which can be captured by measuring immune cell infiltration, a concept that has been strongly correlated with improved survival in multiple cancers [10,11,12]. Thus, as a result of its direct activation of signaling pathways and immune evasion mechanisms, Fn appears to contribute to multiple steps of tumor progression in CRC.

Previous studies indicate that microbial communities may play a role in tumor progression, immune regulation, and treatment response in appendiceal neoplastic syndromes such as pseudomyxoma peritonei (PMP) [13]. In PMP, bacteria have been identified using 16S amplicon sequencing in tumor tissues and mucin deposits, with dominant phyla including Proteobacteria, Actinobacteria, Firmicutes, and Bacteroidetes, as well as the consistent detection of *Helicobacter pylori* using species-specific in situ hybridization [14]. Intra-tumoral microbes have been posited to influence disease progression by driving maladaptive immune responses and directly altering tumor-promoting pathways such as β-catenin signaling [15]. In spite of rapid advances in understanding the microbial contribution to many types of cancer over the past decade, these early findings in AC have not been systematically revisited or extended to other tumor subtypes, such as early-stage or non-mucinous tumors.

As current treatments for advanced AC remain largely ineffective, exploring the impact of intra-tumoral bacteria on tumor biology and immune dynamics is essential for developing more effective therapeutic strategies. The present study was designed to address this knowledge gap by evaluating the Fn density in a cohort of appendiceal cancer specimens, including both primary tumors and peritoneal metastases. Specifically, we aimed to determine the association between Fn density and key clinical, pathological, and immune features, such as tumor grade, peritoneal tumor burden, and immune cell infiltration (CD3+ and CD8+ T cells, M1-/M2-like macrophages) and oncologic outcomes (treatment response and survival).

## 2. Materials and Methods

### 2.1. Patient Cohort Selection

This retrospective study was carried out in accordance with the principles of the Declaration of Helsinki and was approved by the Institutional Review Board of the Allegheny Health Network (AHN) Cancer Institute (protocols 2021-255-AHNMR, 2020-258-AGH and 2022-116-AHNCI-AGH). A waiver of informed consent was granted in accordance with 2018 Common Rule (45 CFR 46), exemption category 4. All patient records and biological samples were de-identified prior to analysis to ensure confidentiality and adherence to ethical standards, and patients were not contacted during the study.

### 2.2. Tissue Handling and Acquisition

Archival, formalin-fixed/paraffin-embedded (FFPE) tissue samples of either primary tumors or metastatic lesions were obtained from 54 unique patients that underwent surgery for appendiceal cancer from March of 2017 to April of 2023. Selection criteria included patients with histologically confirmed AC for whom archival tissue samples and comprehensive clinical follow-up data were available. When the primary tumor was available, the sample was used preferentially for analysis over peritoneal metastatic samples, which were used when no primary tumor specimen was available. Hematoxylin- and eosin-stained slides (4 µm thickness) from each case were selected and reviewed by a board-certified anatomic pathologist to ensure each FFPE block contained representative lesion tissue from each case. Several parallel tissue sections from the selected FFPE block from each case were cut at a thickness of 4 µm and mounted on charged slides for subsequent staining analysis. All stains were performed on the automated BOND RXm stainer (Leica Biosystems, Buffalo Grove, IL, USA).

### 2.3. Compilation of Clinical Records and Clinicopathologic Features

Clinical records and pathologic features of the cohort were extracted from the AHN electronic medical records system, including data on patient demographics (age, sex), clinical variables (tumor marker levels at time of diagnosis, treatment response, recent chemotherapy, history of cytoreductive surgery), tumor characteristics (histologic diagnosis, grade, American Joint Committee on Cancer (AJCC) stage, primary versus metastatic disease status, presence of lymphovascular or perineural invasion) and clinical outcomes (progression-free and overall survival). Survival measures were calculated from the time of analytic tumor sample acquisition to the time of first clinical or radiographic evidence of disease progression or to the time of death. Censoring occurred at the time of the last follow-up. All clinical data was meticulously de-identified to ensure subject privacy.

### 2.4. Detection of Fn Through RNA In Situ Hybridization (RNA-ISH)

To validate detection of Fn utilizing RNA in situ hybridization (RNA-ISH), control samples were tested by injecting Fn organisms into fresh human tissue samples. Fn (ATCC strain 23726) streaked over a sheep’s blood agar plate and incubated in anaerobic conditions until colony growth was observed on the plate. Colonies were suspended in PBS at a density of 8.4 × 10^7^ CFU/mL. Fresh tissue samples were injected with serially diluted Fn aliquots (0.5 mL per 1 cm^3^ of tissue), prior to formalin fixation and paraffin embedding (FFPE). An Fn-negative control sample was injected with PBS alone. Three 3 mm cores from each FFPE sample were collected and a tissue microarray was created containing both the non-injected sample cores and the serially diluted sample cores. The tissue microarray blocks were sectioned to 4 µm thickness and placed on charged slides. RNA-ISH assay was performed using RNAscope™ 2.5 LSx Reagent Kit-RED (Cat 322750, Advanced Cell Diagnostics, Newark, CA, USA) with target probe for Fn (GenBank accession number CP003723.1, Cat. 486418, Advanced Cell Diagnostics). Control staining was also performed on a slide of the tissue microarray using PBS to detect any possible contaminants or artifacts within the staining process. After staining, the slides were dehydrated in a 60 °C oven, followed by coverslipping. The slides were scanned using Aperio Versa 8 (Leica Biosystems) and Fn was visually identified, assessed for individual or clustered Fn forms in a semi-quantitative assay as previously described ([16,17,18,19]; Figure 1). To further mitigate possible false positive identification of Fn clusters, the RNAscope^®^ Control Slide -Human Hela Cell Pellet (Cat. 310045, Advanced Cell Diagnostics) was stained with the same RNA-ISH assay and used as additional negative control to our visual assessment.

### 2.5. Detection of Fn in Clinical Appendiceal Tumor Specimens and Validation of Density Assay

After validating the RNA-ISH assay, we proceeded to assess clinical appendiceal tumor specimens. For each slide from our unique cases, the RNA-ISH assay was performed as described above. Whole-slide images from tumor and adjacent peri-tumoral tissue from each case were scanned using Aperio Versa 8 (Leica Biosystems). Fn was visually identified in each region of interest (Figure 1), and the individual and cluster forms of Fn were manually counted. The volume of tissue examined was calculated from its cross-sectional area multiplied by thickness (4 µm), and the density of Fn was then calculated per unit of volume examined for each case.

To confirm the validity of the semi-quantitative visual density assessment, a quantitative digital imaging algorithm was applied to the digitally scanned slides, analogous to the report by Serna et al. [17]. Using a macro script developed for ImageJ (Version 1.53t), images were converted to stacked images in the Lab color space, with gating thresholds selected manually for red (76–195), green (139–203), and blue (82–130). These images were converted to binary images, which displayed RNA-ISH hybridization signals that could be overlaid onto the original images to establish size-gating rules for individual signals to match the observed morphology of Fn forms and clusters. The script was then applied to a selection of 104 images obtained from a subset of 11 validation cases of appendiceal tissue selected across decile intervals of Fn density in order to ensure a broad range of expected results. Linear regression was then used to confirm a strong correlation between the visual and digital assessment for each case (r = 0.68, *p* = 0.02), as previously reported [20].

### 2.6. Immune Cell Detection Through Immunohistochemistry (IHC), Digital Imagining and Analysis

Using the mounted slides, immunohistochemical staining was used to determine immune cellular densities in our samples. To detect CD3+ and CD8+ T cells, sections were subsequently incubated with ready-to-use primary antibodies: anti-CD3 (LN10, PA0553, Leica Biosystems) and anti-CD8 (4B11, PA0183, Leica Biosystems). M1-like macrophages were identified through co-staining with ready-to-use anti-CD68 (514H12, PA0273, Leica Biosystems) and 1:100 diluted anti-CD86 (E2G8P, 91882S, Cell Signaling Technology, Danvers, MA, USA). M2-like macrophages were identified through co-staining with ready-to-use anti-CD68 (514H12, PA0273, Leica Biosystems) and 1:10,000-diluted anti-CD206 (ab64693, Abcam, Cambridge, UK). Both CD86 and CD206 were diluted using Primary Antibody Diluent (AR9352, Leica Biosystems). All primary antibodies were incubated for 15 min. For detection, a Bond Polymer Refine Detection kit (DS9800, Leica Biosystems) was used for CD3, CD8, and CD206 antibodies. For CD86 detection, BOND Polymer Refine Red Detection (DS9390, Leica Biosystems) was used. CD68 primary antibody used BOND Polymer Refine HRP Plex Detection kit (DS9914, Leica Biosystems) and Green Chromogen (DC9913, Leica Biosystems). After staining, the slides were dehydrated through a series of ethanol and xylene, followed by coverslipping.

Whole-slide images from each case were scanned using Aperio Versa 8 (Leica Biosystems) and analyzed using the eSlide Manager Spectrum Version 12.5.0.6145 (Leica Biosystems), and open-source software, Qupath and ImageJ (QuPath version 5) (Figure 2). The cross-sectional area of examined tissue for each case was then calculated and converted to volume (area x thickness). Areas of acellular mucin were excluded from the analysis. CD3+ and CD8+ T cells and M1- and M2-like macrophages were quantified by counting positive cells within defined tumor and peri-tumoral areas (within 50 μm of neoplastic mucosa) areas after manually tuning the detection algorithm. Cell density was defined as the number of positive cells per cubic millimeter of analytic tissue volume in each section. T cell and macrophage cell densities were then separately converted to percentile scores across all cases, and a composite metric (“I-score”) was derived by averaging these percentiles within each case in a manner analogous to the Immunoscore^TM^, as previously reported [21]. The ratio of cell density was also calculated and recorded for each sample.

### 2.7. Statistical Analysis

The density of Fn in each sample was visually scored, then log-transformed and tested for association with the patient (age, sex) and tumor (grade, peritoneal cancer index (PCI), primary vs metastatic) variables, immune cell densities (CD3/CD8 lymphocytes and tumor-associated M1-like/M2-like macrophages), and oncologic outcome (progression-free and overall survival, PFS/OS). Subgroup comparisons were carried out using non-parametric tests of association (Spearman’s rank correlation test) and linear regression, and survival analysis was performed using Cox regression. To test for agreement between visual and digital density assessment, Pearson’s correlation coefficient was calculated. All statistical analyses were performed using Stata version 18 (StataCorp, College Station, TX, USA), and statistical significance was defined as *p* < 0.05.

## 3. Results

### 3.1. Cohort Description and Distribution of Fn in Tumor Samples

A total of 54 unique patients with appendiceal tumors were studied, representing a diverse array of histologic subtypes and a relatively equal distribution of tumors by grade (Table 1). Fn density was able to be quantified through the RNA-ISH assay (Figure 1), though a non-normal distribution with rightward skew and high-density outliers was observed (*p* < 0.0001 for non-normality, Figure 3). Although a quantitative measure of Fn clusters per unit tissue volume is presented, we regard these findings to be semi-quantitative since intra-tumoral spatial variation in bacterial density is expected to be a significant potential source of random error in descriptive studies [22].

### 3.2. F. nucleatum with Baseline Patient and Tumor Characteristics

We assessed whether Fn density was associated with baseline patient characteristics. Increasing patient age was associated with a high Fn density (Spearman’s r = 0.3, *p* = 0.01; Figure 4). Fn density was not related to biological sex, and our cohort did not have sufficient racial diversity to power a subgroup analysis based on race.

Given the fundamental importance of histologic subtype and grade in AC biology, we next tested these parameters for association with Fn density. We found weak evidence of subgroup variation by histologic diagnosis using the Kruskal–Wallis equality of populations rank test (χ2(4) = 9.3, *p* = 0.05), which was corroborated using one-way ANOVA to model log-transformed Fn density as a function of histologic type (F(4, 45) = 2.64, *p* = 0.046; Figure 5).

An initial assessment of Fn density variation by grade was performed using the aforementioned tests and did not identify significant differences across grades (Kruskal-Wallis χ2(2) = 4.6, *p* = 0.1; one-way ANOVA F(2,50) = 2.57, *p* = 0.09). However, when directly comparing individual grades, we found that intermediate-grade (G2) tumors contained greater Fn density than G1 or G3 tumors (two-sample Wilcoxon rank-sum test z = −2.13, *p* = 0.03; Figure 5). Although based on limited sample size, this finding may hold relevance in the distinct biological characteristics of intermediate-grade tumors, in terms of aggressiveness and propensity for regional or distant metastasis.

Because our tumor population contained both primary lesions and peritoneal metastases, we then tested for variation between these subsets. We did not find a significant difference based on primary vs. metastatic tumors (Kruskal-Wallis χ2(1) = 2.71, *p* = 0.1), although it is notable that all outlier samples were derived from primary tumors (Figure 5).

Fn density was then stratified on the basis of AJCC stage, presence or absence of lymph node metastases, and presence of lymphovascular or perineural invasion. None of these pathologic tumor characteristics was significantly associated with Fn density. To further interrogate the association of overall disease volume with Fn density, we utilized the peritoneal carcinomatosis index and carcinoembryonic antigen levels as surrogate markers. Neither of these indicators were associated with Fn density.

We next hypothesized that treatment history could impact Fn density within AC and tested this hypothesis by stratifying patients based on prior cytoreductive surgery or receipt of chemotherapy within three months prior to tumor tissue resection. Neither of these treatment-related variables was significantly associated with Fn density.

### 3.3. F. nucleatum and Tumor Immune Microenvironment Within AC

Fn density has previously been found to vary with indicators of tumor immune microenvironment in colorectal cancer [23]. We hypothesized that such variation may also occur in AC and tested this hypothesis by measuring the density of lymphocytes (CD3+-and CD8+-T cells) and tumor-associated macrophages (M1-like and M2-like phenotypes) within AC samples (Figure 2). We found a significant negative association between Fn density and overall T lymphocyte infiltration (log-linear regression F(1,46) = 5.56, *p* = 0.02) and CD8+-T lymphocyte infiltration (log-linear regression F(1, 45) = 6.84, *p* = 0.01). Subgroup analysis indicated that these results were limited to high-grade (G3) tumors (Figure 5). Tumor-associated macrophage levels were positively associated with Fn density in low-grade (G1) tumors. This was seen for both M1-like macrophages (F(1,13) = 9.17, *p* < 0.01) and M2-like macrophages (F(1, 18) = 4.94, *p* = 0.04; Figure 6).

### 3.4. F. nucleatum and Oncologic Outcome in AC

Finally, to assess the relationship between AC and oncologic outcomes in AC, we first evaluated whether treatment response to systemic therapy was associated with Fn density by stratifying patients on the basis of best response. We found no difference among defined subsets (no evidence of disease vs. partial response vs. stable disease vs. progressive disease; Jonckheer-Terpstra test for trend, z = −0.56, *p* = 0.6).

We then used an open access web application (Cutoff Finder) to optimize an Fn tissue density cutoff value for prognostic stratification, indicating the 40th percentile as an ideal threshold for analysis [24]. The cutoff value was used for stratification over conventional quartile/median splits given our rightward skew and high-density outliers as well as to dichotomize our observations for this current exploratory study. Stratifying cases at this level, we found an overall survival advantage in patients with high Fn density, with a hazard ratio (HR) of 0.19 (95% CI [0.05, 0.73], *p* = 0.02); Table 2 and Figure 7. We then performed multivariable analysis using grade as a covariable, given the dominant effect of grade on survival in AC [25]. A high Fn density in the latter analysis remained a significant predictor of favorable overall survival independent of tumor grade, with an HR of 0.11 (95% CI [0.02, 0.52], *p* = 0.005); Table 2. Likewise, favorable progression-free survival trended toward statistical significance among high-Fn-density cases HR 0.5 (95% CI [0.2, 1.06], *p* = 0.07), although these prognostic findings are considered exploratory in nature, and should be interpreted with caution given the limited sample size.

## 4. Discussion

The role of Fn in gastrointestinal malignancies has been well established, particularly in CRC, where its presence is associated with tumor progression, immune suppression, and poor outcomes [26,27]. However, little is known about its impact on AC, a rare and distinct malignancy often presenting with peritoneal dissemination. In this study, we evaluated Fn density in AC specimens and explored its associations with tumor characteristics, immune cell infiltration, and clinical outcomes. Not surprisingly, given that Fn is a normal appendiceal flora component in healthy and diseased states [10,28,29,30], we identified Fn forms within all samples, although our semi-quantitative scores varied. In a cohort of diverse histological subtypes of AC, we found a broad range of intra-tumoral Fn density, with a right-skewed distribution and high-density outliers. Intra-tumoral Fn density was weakly associated with—but not specific to—histologic subtype and grade. Moreover, Fn was identified with equal frequency in primary and peritoneal metastatic tumors. Taken together, these results indicated that Fn is a prevalent constituent within AC across all types, stages, and grades.

In colorectal cancer, substantial variation in the intra-tumoral microbiome has been reported on the basis of patient factors such as age, metabolic variables, diet, body mass index, and antibiotic use. For example, a recent 16s rRNA amplicon sequencing analysis of CRC cases determined that early-onset cases had significantly higher α and β diversity and *Akkermansia* and *Bacteroides* density relative to average-age-onset CRC patients, whose tumors contained greater amounts of *Bacillus*, *Staphylococcus*, *Listeria*, *Enterococcus*, *Pseudomonas*, *Fusobacterium*, and *Escherichia*/*Shigella* [31]. Similarly, in a meta-analysis of fecal metagenomics studies, a 40% increase in the odds ratio of Fn positivity was seen per decade of increased age in CRC patients [32]. We found a similar trend—i.e., a positive association between Fn and patient age—in AC. Other studies in CRC have noted inverse associations between Fn and dietary influences such as dairy and fiber intake [33,34]. Because AC is rare and sporadic, very little is known about age- or diet-related influences on AC or its microbiome.

Species-specific probes and culture methods have highlighted the presence of *H. pylori* and other target organisms in PMP samples, but the presence of Fn within AC or PMP has not been previously described to our knowledge. This is in spite of ample evidence that Fn is a frequent commensal inhabitant of the healthy appendix [10] and a consistent component of the microbial population in acute appendicitis [28,29,30]. Prior studies have suggested that microbial factors may influence peritoneal disease progression in AC by stimulating oncogenic signaling pathways and mucin production. Although a mechanistic or causal link between intra-tumoral bacteria and AC disease progression is far from certain at this point, it has been hypothesized that antibiotic use in PMP patients could be an effective strategy for disease control when combined with conventional modalities such as cytoreductive surgery or intra-peritoneal chemotherapy [15,35].

While the previous literature on microbiome in AC is scant, much more is known about the microbial content of the healthy appendix and in acute appendicitis. The appendiceal microbiome has been shown to be highly variable among patients [36], to be distinct in composition from the rest of the gastrointestinal tract [10], to be resistant to luminal conditions such as illness or antibiotic use [37], and is theorized to serve as a ‘inoculum’ for repopulation of the hindgut following disruption due to disease [10]. In normal development, the translocation of luminal microbes across the appendiceal mucosa is integral to the development of the gut-associated lymphoid tissue and intestinal mucosal immunity to pathogens and tolerance of commensal organisms [38]. Notably, Fn outgrowth is commonly associated with acute appendicitis, especially in children [11,39,40]. Moreover, appendectomies may influence the microbial composition of the colon, including the potential enrichment of cancer-associated taxa and diminishment of taxa thought to be protective against CRC [41]. On the other hand, prior appendectomy was found to be protective against Fn-associated CRC [12]. Clearly, while we have much to learn regarding the impact of the AC microbiome on disease development and progression, it is clear that the appendiceal microbiome is dynamic and carries wide ranging impacts on tumor biology both in the appendix and in the adjacent colon.

Our findings suggest a unique relationship between the tumor immune microenvironment and Fn in AC. Prior studies in CRC have demonstrated that Fn can suppress immune responses, enabling immune escape and tumor progression [42]. We found an overall negative association between lymphocyte and Fn densities, which is in parallel with previously reported studies in CRC highlighting lymphocyte apoptosis activation by Fn-derived Fap2 and RadD [43]. We also noted an overall positive association between TAM and Fn density in AC. Moreover, we found preliminary evidence that these associations are histologic grade-dependent, although further corroboration in larger datasets will be necessary. We have previously reported that intra-tumoral lymphocyte density is associated with improved survival in AC [44,45], whereas the M2-like macrophage density is associated with higher disease volume and progression risk in AC [46]. These apparent prognostic implications of the immune microenvironment in AC make any association with microbiome influences noteworthy and deserving of additional investigation.

Our findings suggest that the microbiome’s influences on AC cancer may be distinct from those in CRC. Interestingly, the association of specific leukocyte subtype densities in CRC with tissue Fn content has identified negative association with overall T lymphocyte (CD3+) density and memory-type helper T cell density (CD3+CD4+CD45RO+), whereas CD8+ lymphocyte density and M1-like and M2-like macrophage densities were not associated with Fn content [23]. Macrophage polarization to the M2 phenotype has been induced in vitro by Fn via activation of the TLR4/KF-κB/S100A9 cascade [47,48,49,50,51], although these findings have not been demonstrated in vivo [23]. The increase in M1-like macrophages in AC could hypothetically be attributed to unique microbiota influences within the appendix. The structural components and metabolites of microbes can activate specific pattern recognition receptors on macrophages, thereby guiding differentiation pathways into M1-like, M2-like, or other phenotypes. Given the differences in the microbial constituents between AC and CRC, much remains to be learned regarding the specific contributions of microbial elements in appendiceal tumorigenesis, which cannot be assumed to mirror those in CRC.

We found an association between a high Fn content and improved survival in AC. Strikingly, this association was apparent even in a modest cohort of patients and when controlled for histologic grade—the most consistent and reliable conventional prognostic factor in AC [25]. In theory, Fn could have a protective biological influence on AC, or could simply be a passenger organisms tied to a less aggressive subset of tumors—possibilities that will require further study to untangle. Our finding that Fn content correlates with improved survival contradicts the existing CRC literature, though the reasoning behind this remains elusive as our study lacks a functional assay to speculate immunomodulatory effects of Fn and our improved survival. Regardless of causality, this result lends potential utility to intra-tumoral Fn density as a potential prognostic biomarker in AC, either as a standalone variable or in combination with other indicators of prognosis. Once again, these results differ markedly from those in the existing CRC literature, where Fn density has been repeatedly identified as a poor prognostic factor [16,17,52,53].

This study has important limitations, including a retrospective design and modest sample size, which may limit its statistical power, particularly in subgroup comparisons involving survival analysis. In this study, we were unable to assess important mechanistic questions regarding the mechanisms by which Fn may contribute to appendiceal tumor growth and dissemination. Fn encodes several adhesins for interspecies interactions, including Fap2, RadD, aid1, and FadA [43,54,55]. FadA, the best-characterized virulence factor for Fn [56], is the only adhesin found to bind host cells [57]. FadA is also an invasin, required for binding and invasion of both normal and cancerous cells [58]. Future studies are planned to assess the specific role of Fn virulence factors in promoting tumorigenesis in AC, as well as in modulating immune cell infiltration and oncologic outcomes. Regarding our assessment of the immune microenvironment in AC, the present analysis of lymphocyte and macrophage densities are rudimentary assays that will be followed by more detailed multiplex spatial assays that are capable of generating a comprehensive picture of the immune contexture within AC. Methodologic constraints of quantifying bacterial density in archival tissue are fully acknowledged and include the risk of substantial intra-tumoral heterogeneity and loss of bacterial forms due to tissue processing techniques. We have mitigated these limitations by examining entire cross-sectional images of individual tumor blocks from each case, by adhering to techniques identical to those in high-impact publications on CRC [16,17,22], and by internally validating our RNA-ISH assay and semi-quantitative visual assessment with digital image analysis software, as previously described [17].

Our study alludes to the possible immunomodulatory roles of Fn, though as mentioned, our study lacks a functional assay. We fully acknowledge this limitation though the scope of this study was to first detect and quantify Fn in AC samples and we plan to carry out functional assessments to fill this knowledge gap. Prospective studies in larger cohorts, using orthogonal techniques (such as 16S rRNA amplicon sequencing or metagenomics methods) to assess the tumor microbiome are urgently warranted to further explore the role of Fn and other organisms in AC pathogenesis. An in depth examination into AC TME, especially a functional analysis, of Fn is warranted prior to making therapeutic decisions, such as the potential use of microbiome-targeted interventions. We hypothesize that antimicrobial therapy may be a potential therapeutic option for some patients, but given the survival benefit observed in our AC population, anti-Fn directed therapies may not be efficacious. Although targeted therapy directed towards the immune modulatory effects of Fn or antibiotic treatment directed at other microbiota may be effective, this will need to be determined through further in-depth analysis into the TME of AC with validation via in vivo or ex vivo studies.

## 5. Conclusions

In conclusion, this study demonstrates that *F. nucleatum* is prevalent in AC and is associated with patient age, tumor grade, and stage, reduced intra-tumoral T-cell and increased TAM density, and favorable prognosis in AC. These associations suggest the possibility of a formative influence of tumor microbiome on AC biology that will require careful mechanistic follow-up investigation and raise potentially important differences between AC and CRC, where distinct and/or opposite trends have been observed.

## Figures and Tables

**Figure 1 microorganisms-13-01644-f001:**
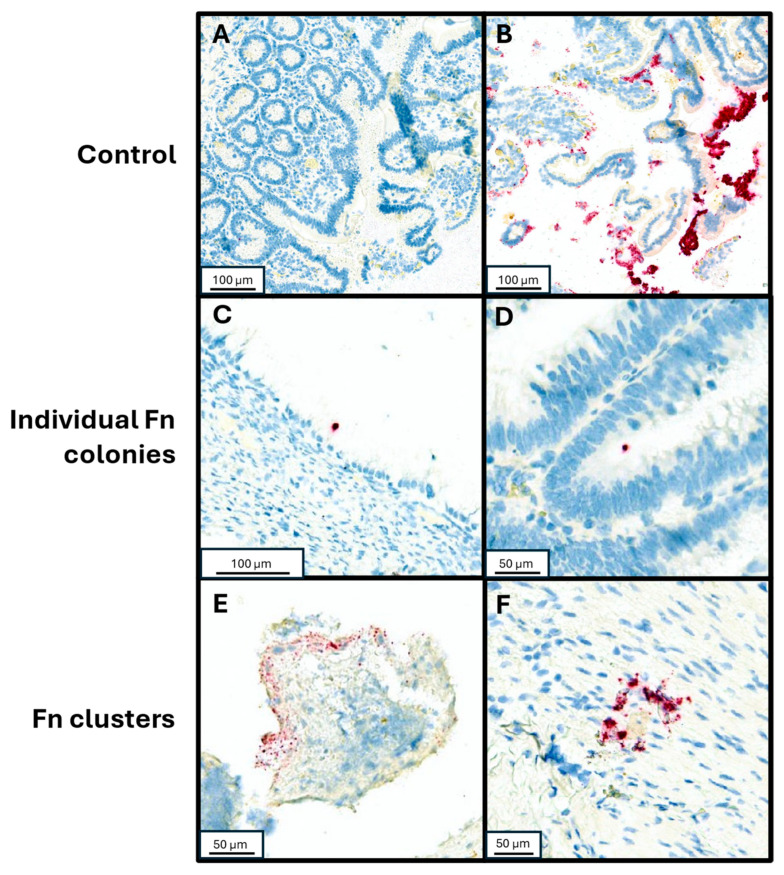
RNA-in situ hybridization with Fn-specific probe. (**A**) Negative control specimen of small bowel tissue injected with PBS. (**B**) Positive control of small bowel tissue injected with undiluted resuspended Fn culture. (**C**,**D**) Individual Fn cells adherent to neoplastic mucosal tissue from patients with mucinous adenocarcinoma and low-grade appendiceal mucinous neoplasm, respectively. (**E**,**F**) Clustered Fn forms from patients with high grade goblet cell adenocarcinoma and a premalignant serrated sessile lesion of the appendix, respectively.

**Figure 2 microorganisms-13-01644-f002:**
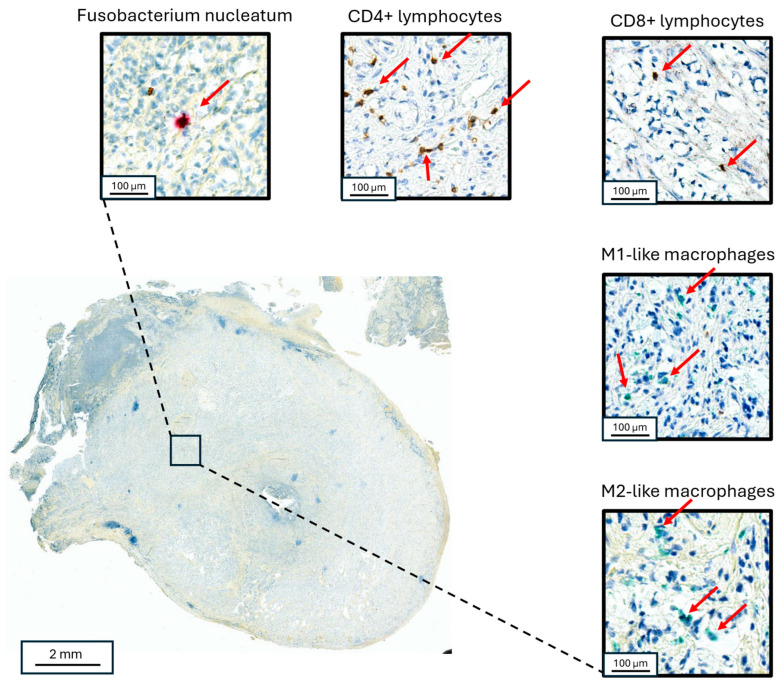
Co-localized staining of Fn and immune cells in appendiceal neoplasms. Staining for Fn (**upper left**), CD3+ lymphocytes (**upper middle**), CD8+ lymphocytes (**upper right**), M1-like macrophages (**right middle**) and M2-like macrophages (**lower right**) in a goblet cell adenocarcinoma primary appendiceal tumor. The red arrows point to the identified target cells in each panel, respectively.

**Figure 3 microorganisms-13-01644-f003:**
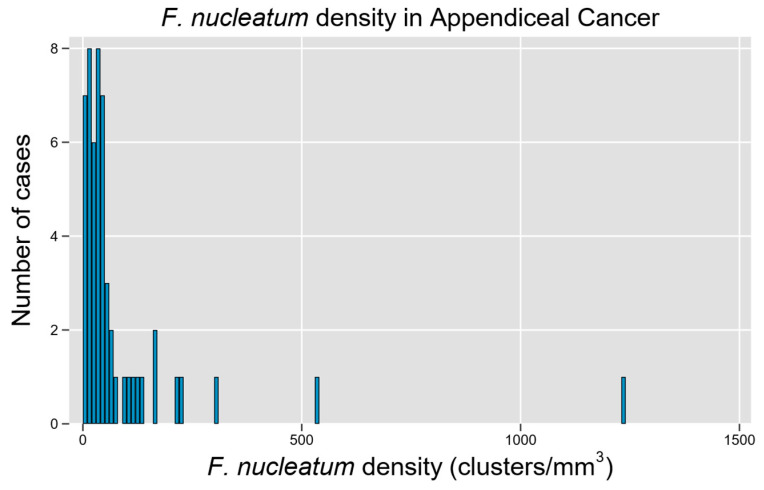
*F. nucleatum* is prevalent and widely variable in tissue tumor content among cases of appendiceal cancer. Histogram of Fn density among examined cases demonstrates a right-skewed distribution with high-Fn-density outliers.

**Figure 4 microorganisms-13-01644-f004:**
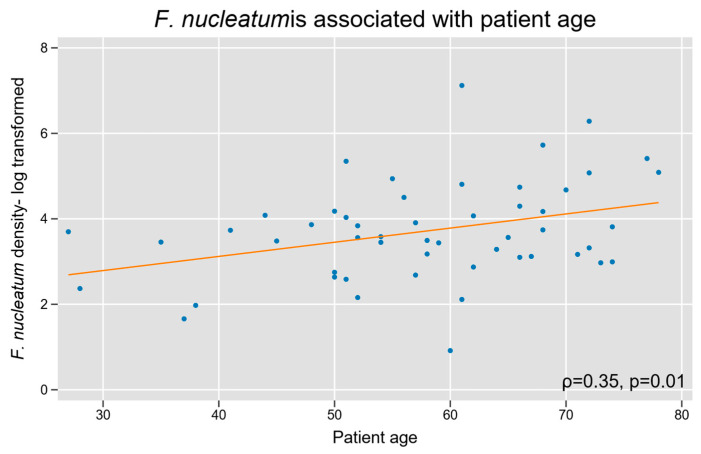
Intra-tumoral Fn density and patient age in appendiceal cancer. A log-linear relationship was observed between tumor tissue Fn content and increasing age.

**Figure 5 microorganisms-13-01644-f005:**
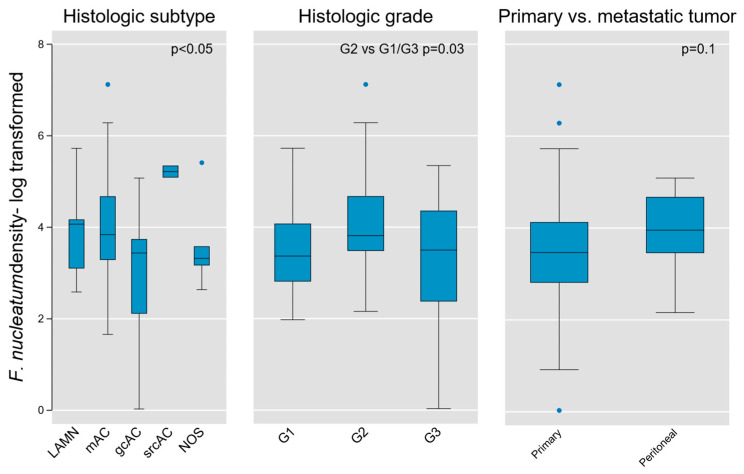
Association of intra-tumoral Fn density with tumor histologic type, grade, and location (primary vs. peritoneal metastases) in AC. Weak evidence for variation on the basis of histologic type and grade was found (*p* value shown for Kruskal–Wallis equality of populations rank test), whereas no measurable difference was obtained between primary and metastatic lesions. LAMN, low grade appendiceal mucinous neoplasm; mAC, mucinous adenocarcinoma; gcAC, goblet cell adenocarcinoma; srcAC, signet ring cell adenocarcinoma; AC NOS, adenocarcinoma not otherwise specified; G1, Grade 1; G2, Grade 2; G3, Grade 3.

**Figure 6 microorganisms-13-01644-f006:**
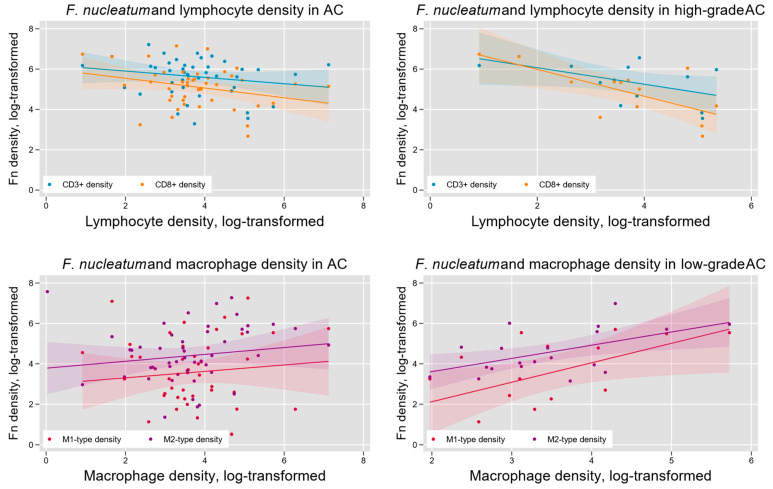
Intra-tumoral Fn density and the AC immune microenvironment. Overall, tumor Fn content was negatively associated with intra-tumoral lymphocyte density (**upper left**) and positively associated with tumor-associated macrophage (TAM) density (**lower left**). These relationships were most pronounced in high-grade tumors for lymphocytes (**upper right**) and low-grade tumors for TAMs (**lower right**).

**Figure 7 microorganisms-13-01644-f007:**
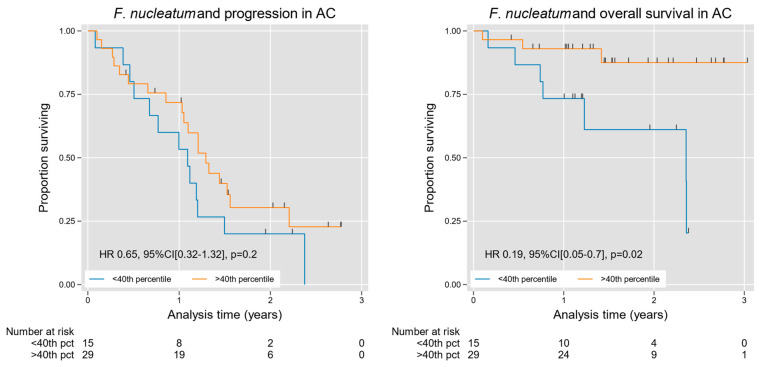
Kaplan–Meier curves for progression and survival in AC, stratified by intra-tumoral Fn density. Fn was significantly associated with improved survival in AC (Table 2), a finding that was grade independent. A trend toward improved progression-free survival was also observed on multivariable analysis (Table 2). For progression-free and overall survival analysis, 49 patients had outcome metrics available at the time of analysis. On each graph, the blue line and the orange line represent the <40th percentile and the >40th percentile groups, respectively, determined by the CuttOff Finder as described by Budczies et al [24].

**Table 1 microorganisms-13-01644-t001:** Patient cohort characteristics. LAMN, low grade appendiceal mucinous neoplasm; mAC, mucinous adenocarcinoma; gcAC, goblet cell adenocarcinoma; srcAC, signet ring cell adenocarcinoma; AC NOS, adenocarcinoma not otherwise specified; AJCC, American Joint Committee on Cancer; NED, no evidence of disease; PCI, peritoneal cancer index; CEA, carcinoembryonic antigen; n/a, not applicable. * only data available reported.

Characteristic		Total Patients n = 54
age	Median (IQR)	59 years old (51–68)
sex	Male	46.3% (*n* = 25)
	Female	53.7% (*n* = 29)
histology	LAMN	18.5% (*n* = 10)
	mAC	31.5% (*n* = 17)
	gcAC	20.3% (*n* = 11)
	srcAC	7.4% (*n* = 4)
	AC NOS	18.5% (*n* = 10)
	Villous adenoma	1.9% (*n* = 1)
	Sessile serrated lesion	1.9% (*n* = 1)
tumor site	Primary appendiceal	74.1% (*n* = 40)
	Metastatic lesion	25.9% (*n* = 14)
grade	n/a	3.7% (*n* = 2)
	1	37.0% (*n* = 20)
	2	29.6% (*n* = 16)
	3	29.6% (*n* = 16)
stage (AJCC)	Non-invasive	13.0% (*n* = 7)
	I	1.9% (*n =* 1)
	II	20.4% (*n* = 11)
	III	5.6% (*n* = 3)
	IV	59.3% (*n* = 32)
nodal status *	Positive	51.4% (*n* = 18)
	Negative	48.6% (*n* = 17)
lymphovascular invasion *	Positive	23.5% (*n* = 8)
	Negative	76.5% (*n* = 26)
perineural invasion *	Positive	39.3% (*n* = 11)
	Negative	60.7% (*n* = 17)
recent chemotherapy	Within 3 months	21.6% (*n* = 11)
prior cytoreductive surgery	Yes	15.7% (*n* = 8)
best response to therapy	NED	40.7% (*n* = 22)
	Partial response	24.1% (*n* = 13)
	Stable disease	1.9% (*n* = 1)
	Progression	33.3% (*n* = 18)
PCI	Median (IQR)	26 (18–31)
CEA	Median (IQR)	3.0 (1.3–11.1)

**Table 2 microorganisms-13-01644-t002:** Association of *F. nucleatum* content with tumor progression and survival in AC. Fn, *Fusobacterium nucleatum*; HR, hazard ratio; CI, confidence interval.

			Univariable			Multivariable		
			HR	95% CI	*p*	HR	95% CI	*p*
Progression								
	Fn density							
		<40th percentile	Ref.			Ref.		
		>40th percentile	0.65	0.32, 1.32	0.24	0.46	0.2, 1.1	0.07
	Grade							
		I				Ref.		
		II				2.4	0.9, 6.4	0.07
		III				1.5	0.6, 3.6	0.4
Survival								
	Fn density							
		<40th percentile	Ref.					
		>40th percentile	0.19	0.05, 0.73	0.02	0.11	0.02, 0.52	0.005
	Grade							
		I				Ref.		
		II				10.6	1.05, 107.7	0.05
		III				5.3	0.58, 48.6	0.14

## Data Availability

The raw data supporting the conclusions of this article will be made available by the authors on request.

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
