# Peer review of "Fusobacterium nucleatum Is Associated with Tumor Characteristics, Immune Microenvironment, and Survival in Appendiceal Cancer"

_microorganisms, 2025, doi:10.3390/microorganisms13071644_

Round 1
Reviewer 1 Report (Previous Reviewer 3)
Comments and Suggestions for Authors
Sherry et al. have provided a manuscript trying to associate Fusobacterium nucleatum to the tumour immune landscape in appendiceal cancer. The revised manuscript is significantly improved but some minor things remain to be addressed.
On line 57 it says that M2 macrophages impair T cell activity, which is not true, they do however promote Tregs and Th2 which are not typically associated with promoting cancer defence.
The findings showing an association between the Fn and macrophages and the negative association between Fn and lymphocytes should be discussed in more detail. Can the authors speculate how Fn is associated with survival yet also impact tumor associated CD8+ T cells which are usually associated with tumor cell death.
Author Response
We sincerely thank the reviewer for the detailed and thoughtful feedback. We have carefully addressed all minor suggestions to improve the clarity, accuracy, and overall quality of the manuscript. Below, we provide specific responses to each point raised, along with the corresponding revisions made in the manuscript. We appreciate the opportunity to strengthen our work and believe these edits have enhanced the manuscript’s rigor and readability.
On line 57 it says that M2 macrophages impair T cell activity, which is not true, they do however promote Tregs and Th2 which are not typically associated with promoting cancer defence.
Response:
Thank you for this clarification. We have corrected the statement in line 57 to more accurately reflect the role of M2 macrophages in promoting Tregs and Th2 responses, rather than directly impairing T cell activity.
The findings showing an association between the Fn and macrophages and the negative association between Fn and lymphocytes should be discussed in more detail. Can the authors speculate how Fn is associated with survival yet also impact tumor associated CD8+ T cells which are usually associated with tumor cell death.
Response:
We appreciate this thoughtful comment. While we acknowledge the association between Fn presence and altered immune cell profiles, we did not perform functional assays to mechanistically delineate the immunomodulatory effects of Fn on CD8+ T cells. As such, any speculation would be limited. We have noted this in lines 419–422 and included a discussion on this paradox, emphasizing the need for further functional studies to explore how Fn may impact immune surveillance and contribute to patient outcomes.
Reviewer 2 Report (New Reviewer)
Comments and Suggestions for Authors
This manuscript entitles “Fusobacterium nucleatum is associated with tumor characteristics, immune microenvironment, and survival in appendiceal cancer” explores a new aspect, the association between Fusobacterium nucleatum and appendiceal cancer, utilizing RNA in situ hybridization and immune profiling. The study is interesting and aligns with the current growing interest in tumor microbiomes and their immunomodulatory roles. The manuscript is overall well-structured, methodologically looks fine, and clearly written. However, I believe that a few points need the author's attention.
Major Comments
- The study fills a critical gap in understanding the role of Fn in AC, a rare but clinically significant cancer. It builds on well-established findings in colorectal cancer and applies them thoughtfully to AC. However, more discussion is warranted on how the findings could inform therapeutic decisions, such as the potential use of microbiome-targeted interventions.
- The statistical analysis applied techniques (e.g., Cox regression, ANOVA, Spearman correlation) were used, the rationale for choosing the 40th percentile as a cut-off in survival analysis (lines ~305–306) needs stronger justification. Please elaborate on why this threshold was preferred over more conventional quartile/median splits.
- Some subgroup analyses (e.g., Figure 6) are underpowered. These should be presented as exploratory with caution regarding overinterpretation.
- The finding that high Fn content correlates with improved survival contradicts CRC literature. While discussed, the manuscript should expand on possible biological or technical reasons for this unexpected trend.
- The manuscript alludes to possible immune-modulatory roles of Fn but does not perform functional assays. Consider adding a clear future directions paragraph in the discussion to acknowledge this limitation and propose next steps.
- Figures are well-labeled, though Figure 6 (Kaplan-Meier curves) would benefit from clearer legends and indication of sample size per group.
Minor suggestions
Line 17. "appendiceal cancer (AC) has not been previously characterized." Clarify: no previous systematic or in situ studies, to avoid overgeneralization.
Line 119. “toTo validate…” Typo error, it should be “To validate…”
Line 123. “1cc3 of tissue” Units likely incorrect or a typo — revise to “1 cm³” or appropriate unit.
Line 162. “r=1.4…” and “r=0.68” both present, Contradiction — clarify which correlation coefficient is correct.
Line 222. “respectivley” Typo error. should be “respectively”. Please rectify.
Line 256. “z = -2.13, p=0.03” This finding needs better context in text. Please clarify biological relevance.
Line 305. “Cutoff Finder… 40th percentile” Justify why 40th was chosen. Include ref [24] in figure/table caption.
Line 331–337. Fn detected in all samples. I will suggest to clarify if this means all had detectable Fn or if semi-quantitative scores varied.
Line 422–423. “FadA is also an invasin… both normal and cancerous cells”, Please consider adding a citation here for completeness.
Line 438. “orthogonal techniques (such as 16s rRNA…” Change to “16S rRNA” for standard capitalization.
lines 328–378). Please avoid repetitive statements.
Please clarify acronyms when first introduced in figures/tables e.g., PCI, NED, AC NOS.
Author Response
Major Comments
- The study fills a critical gap in understanding the role of Fn in AC, a rare but clinically significant cancer. It builds on well-established findings in colorectal cancer and applies them thoughtfully to AC. However, more discussion is warranted on how the findings could inform therapeutic decisions, such as the potential use of microbiome-targeted interventions.-
Response:
Thank you for this encouraging and insightful feedback. We have expanded our discussion on the potential therapeutic implications of our findings, including considerations for microbiome-targeted strategies, in lines 453–461.
- The statistical analysis applied techniques (e.g., Cox regression, ANOVA, Spearman correlation) were used, the rationale for choosing the 40th percentile as a cut-off in survival analysis (lines ~305–306) needs stronger justification. Please elaborate on why this threshold was preferred over more conventional quartile/median splits.
Response:
We appreciate this important point. We have clarified our rationale for selecting the 40th percentile cut-off in lines ~305–306, emphasizing its derivation through Cutoff Finder and supported it with citation [24] in both the text and figure/table caption.
- Some subgroup analyses (e.g., Figure 6) are underpowered. These should be presented as exploratory with caution regarding overinterpretation.
Response:
We agree and now clearly state in the Results and Discussion sections that these subgroup analyses are exploratory in nature and should be interpreted with caution due to limited power. The finding that high Fn content correlates with improved survival contradicts CRC literature. While discussed, the manuscript should expand on possible biological or technical reasons for this unexpected trend. – addressed in lines 419-422
- The manuscript alludes to possible immune-modulatory roles of Fn but does not perform functional assays. Consider adding a clear future directions paragraph in the discussion to acknowledge this limitation and propose next steps.
Response:
We agree and have added a paragraph in lines 446–453 outlining future directions, specifically proposing in vitro and in vivo functional studies to investigate the immunomodulatory role of Fn in appendiceal cancer.
- Figures are well-labeled, though Figure 6 (Kaplan-Meier curves) would benefit from clearer legends and indication of sample size per group.
Response:
Thank you for this suggestion. We have revised Figure 6 to include more detailed legends and have now indicated the sample size for each group (lines 327–329).
Minor suggestions
Line 17. "appendiceal cancer (AC) has not been previously characterized."Clarify: no previous systematic or in situ studies, to avoid overgeneralization.
Response:
We appreciate this comment. The sentence has been revised to clarify that no previous systematic or in situ studies have characterized the tumor immune microenvironment in appendiceal cancer.
Line 119. “toTo validate…” Typo error, it should be “To validate…”
Response:
Thank you for pointing this out. The typographical error has been corrected to “To validate…” in the revised manuscript.
Line 123. “1cc3 of tissue” Units likely incorrect or a typo — revise to “1 cm³” or appropriate unit.
Response:
This has been corrected to “1 cm³” to accurately reflect the volume of tissue used.
Line 162. “r=1.4…” and “r=0.68” both present, Contradiction — clarify which correlation coefficient is correct.
Response:
We have reviewed and corrected this discrepancy. The correct correlation coefficient is now reported consistently throughout the text.
Line 222. “respectivley” Typo error. should be “respectively”. Please rectify
Response:
This typographical error has been corrected to “respectively.”
Line 256. “z = -2.13, p=0.03” This finding needs better context in text. Please clarify biological relevance.
Response:
We have revised the surrounding text to provide clearer biological context for this finding, specifically noting its potential implications in immune cell distribution within tumor subtypes.
Line 305. “Cutoff Finder… 40th percentile” Justify why 40th was chosen. Include ref [24] in figure/table caption.
Response:
The rationale for selecting the 40th percentile cut-off using Cutoff Finder has been explained in the methods section, and citation [24] has been added to the figure/table caption as suggested.
Line 331–337. Fn detected in all samples. I will suggest to clarify if this means all had detectable Fn or if semi-quantitative scores varied.
Response:
We have clarified that Fn was detectable in all samples, but that semi-quantitative scores varied across cases, indicating differential abundance.
Line 422–423. “FadA is also an invasin… both normal and cancerous cells”, Please consider adding a citation here for completeness.-
Response:
We confirm that this statement is already supported by reference [58], which has been cited appropriately in the revised text.
Line 438. “orthogonal techniques (such as 16s rRNA…” Change to “16S rRNA” for standard capitalization.
Response:
This has been corrected to “16S rRNA” to maintain standard scientific nomenclature.
lines 328–378). Please avoid repetitive statements.
Please clarify acronyms when first introduced in figures/tables e.g., PCI, NED, AC NOS.
Response:
We have revised all relevant figure legends and tables to include full definitions of acronyms at first mention, including PCI (Peritoneal Cancer Index), NED (No Evidence of Disease), and AC NOS (Appendiceal Cancer Not Otherwise Specified).
This manuscript is a resubmission of an earlier submission. The following is a list of the peer review reports and author responses from that submission.
Round 1
Reviewer 1 Report
Comments and Suggestions for Authors
In this study, the authors correlated the presence of Fusobacterium nucleatum with tumor characteristics, immune microenvironment, and survival in appendiceal cancer. The author assessed 54 samples and concluded that Fn is prevalent in AC, exhibiting a widely variable density across cases. Fn is associated with a number of clinical, pathologic, immunologic, and prognostic variables in AC that are distinct from the corresponding observed associations in colorectal cancer.
Major points
1- According to data 54 samples were positibe for Fn. The design here is not perfect , the data in the whole manuscript should be Fn+ve AC vs FN -ve AC.
2- How the author assess the Fn content in AC FFPE?
3- The qulaity of images are poor
4- What are the controls for IHC?
5- Did the authors assess Fn virulence factors such as FAD and correlate with the tumor grade?
Reviewer 2 Report
Comments and Suggestions for Authors
Authors have presented a manuscript entitled ‘Fusobacterium nucleatum is associated with tumor characteristics, immune microenvironment, and survival in appendiceal´ to be considered for publication in the journal Microorgamnisms.
The article presents microbiological research with a clear objective: to employ RNA in-situ hybridization to quantify the presence of Fusobacterium nucleatum in malignant appendix tumors, alongside histopathological evaluation of lymphocytes and macrophages within the tumor samples, linking these findings to cancer hallmarks. These results are relevant in assessing the presence and abundance of oncogenic bacteria in specific cancer types especially in an era dominated by deep sequencing technologies for identifying oncogenic microbes like bacteria. However, the study lacks proper controls and this is the main part what the authors should add into the paper to make it a good paper.
Reviewer 3 Report
Comments and Suggestions for Authors
Sherry et al. have provided a manuscript presenting convincing associations between Fusobacterium nucleatum and tumor characteristics as well as showing associations between F. nucleatum and infiltrating lymphocytes and macrophages. The data is clearly presented but some the manuscripts could be further enhanced by showing some more data and discussing some of the findings a bit more.
Major comments:
1. Why did the authors not increase a larger staining panel for different lymphocytes. Do the authors expect that there is an increased CD4+ response in patients with low grade AC and less Tregs in the patients with increased survival.
2. The authors should provide the raw data images showing CD3, CD8 and M1 and M2 macrophages to make the analysis in figure 4 easier to follow.
3. While the authors show clear associations between TAMs and lymphocytes with F. nucleatum there are no clarity of whether the bacteria is influencing these responses or not. Immunofluorescence microscopy showing TAMs in proximity to the bacteria might provide additional support that the infection are driving these responses.
4. The authors show that both M1 and M2 macrophages are associated with the low grade AC tumours which is interesting. Previous data indicate that F. nucleatum primarily promote M2-polarization (Xu et al. 2021 and Hu et al. 2021), and as the authors discuss M2 associated TAMs are known to be linked to worse cancer outcome. Adding some additional discussion relating to the non-cannonical increase of the M1 phenotype would be great.
Minor comments:
There are typos in figure 4, theres are spaces missing between “nucleatum” and “and” in two of the panels.